# Glycaemia Fluctuations Improvement in Old-Age Prediabetic Subjects Consuming a Quinoa-Based Diet: A Pilot Study

**DOI:** 10.3390/nu14112331

**Published:** 2022-06-01

**Authors:** Diana A. Díaz-Rizzolo, Nihan Acar-Denizli, Belchin Kostov, Elena Roura, Antoni Sisó-Almirall, Pedro Delicado, Ramon Gomis

**Affiliations:** 1Faculty of Health Science, Universitat Oberta de Catalunya, 08018 Barcelona, Spain; ramon.gomis@idibaps.org; 2Primary Healthcare Transversal Research Group, IDIBAPS, 08036 Barcelona, Spain; badriyan@clinic.cat (B.K.); asiso@clinic.cat (A.S.-A.); 3Department of Statistics and Operation Research, Universitat Politecnica de Catalunya, 08034 Barcelona, Spain; acrnihan@gmail.com (N.A.-D.); pedro.delicado@upc.edu (P.D.); 4Primary Care Center Les Corts, CAPSBE, 08036 Barcelona, Spain; 5Department of Health and Alimentary Habits, Alicia-elBulli Foundation, Sant Fruitós de Bages, 08272 Barcelona, Spain; elena@alicia.cat; 6Diabetes and Obesity Research Laboratory, IDIBAPS–Hospital Clinic of Barcelona, 08036 Barcelona, Spain; 7Centro de Investigación Biomédica en Red de Diabetes y Enfermedades Metabólicas Asociadas (CIBERDEM), 28029 Madrid, Spain; 8Department of Endocrinology and Nutrition, Hospital Clinic of Barcelona, 08036 Barcelona, Spain

**Keywords:** nutrition, diabetes, quinoa, diet, prevention, glycaemia

## Abstract

This study aimed to observe if quinoa could produce a benefit on postprandial glycemia that would result in less progression to type 2 diabetes (T2D). A cross-over design pilot clinical study with a nutritional intervention for 8 weeks was performed: 4 weeks on a regular diet (RD) and 4 weeks on a quinoa diet (QD). Nine subjects aged ≥65 years with prediabetes were monitored during the first 4 weeks of RD with daily dietary records and FreeStyle Libre^®^. Subsequently, participants started the QD, where quinoa and 100% quinoa-based products replaced foods rich in complex carbohydrates that they had consumed in the first 4 weeks of RD. The glycemic measurements recorded by the sensors were considered as functions of time, and the effects of nutrients consumed at the intended time period were analyzed by means of a function-on-scalar regression (fosr) model. With QD participants, decreased body weight (−1.6 kg, *p* = 0.008), BMI (−0.6 kg/m^2^*p* = 0.004) and waist circumference (−1.5 cm, *p* = 0.015) were observed. Nutrients intake changed during QD, namely, decreased carbohydrates (*p* = 0.004) and increased lipids (*p* = 0.004) and some amino acids (*p* < 0.05). The fosr model showed a reduction in postprandial glycemia in QD despite intrapersonal differences thanks to the joint action of different nutrients and the suppression of others consumed on a regular diet. We conclude that in an old age and high T2D-risk population, a diet rich in quinoa reduces postprandial glycemia and could be a promising T2D-preventive strategy.

## 1. Introduction

Quinoa is a pseudocereal and has potential health benefits and exceptional nutritional value: a high concentration of protein with all essential amino acids, unsaturated fatty acids (FA), vitamins and minerals [1]. Furthermore, quinoa is rich in fiber and complex carbohydrates both making it a low glycemic index grain [1]. Among other nutrients, phenolic compounds found in quinoa [2,3] show inhibitory effects on α-glucosidase and lipase activities [4,5] that are involved in sugar and lipid digestion in the digestive tract, respectively. Specifically, it has recently been observed that the polyphenols present in quinoa could have an effect on the reduction in postprandial blood glucose [6]. All this leads us to think that quinoa can be used to improve the profile of metabolic risk factors and help control type 2 diabetes (T2D) [3,4,7].

T2D is, along with its complications, a major cause of premature death [8,9]. Prediabetes (preDM) describes that up to 70% of people in this state will develop T2D, with an annual rate of around 10% [10,11]. Nevertheless, the number of new T2D onsets in ≥65-year-olds stands out in particular, increasing by 4.5 compared to 3 times the conversion ratio in the total population [12].

Despite the multiple bioactive components presented by this grain and the multiple biological activities associated with them [13], few studies exist on quinoa in vitro, in vivo and in clinical trials, for assessing its potential clinical applications supported by strong scientific evidence [1]. Notably, there is a gap in well-designed T2D studies, and, more specifically, there are no studies related to the prevention of this disease. Thus, there is a need for increased scientific research in this field.

Glucose sensors measure concentrations over a long time period and record data for specific time intervals by repeated periods. This type of longitudinal data can be handled as a curve. A recent study [14] that analyzed longitudinal data on glucose concentrations by means of classical statistical methods concluded that consumption of 20 g quinoa per day showed a reduction in postprandial glycemic response. However, recent statistical methods, such as functional data analysis (FDA; see for instance Ramsay and Silverman, [15]), have been proposed to analyze the data considering their continuous variation over time. An FDA approach would allow the glucose concentrations provided by each participant to be handled as a function of time.

Due to the nutritional properties of quinoa and the limited information from studies conducted on this grain, especially in T2D prevention, we hypothesize that replacing food that contributes large amounts of complex carbohydrates to the diet with quinoa will produce a T2D preventive impact in subjects at high risk of developing it. Therefore, the aim of the current study was to evaluate the effect of cereal, flour, tuber and legume replacement for the same amounts of quinoa and quinoa products, on blood glucose fluctuations as a functional linear model in old-age with preDM subjects.

## 2. Methods

This study was a pilot clinical trial according to a cross-over design developed in IDIBAPS-Hospital Clinic de Barcelona with registration number NCT04529317 and date of registration 27 August 2020 in ClinicalTrials.org.

### 2.1. Participants

Subjects included, from whom informed consent was obtained for experimentation, and study was performed in accordance with The Code of Ethics of the World Medical Association, were males and females ≥65 years old with fasted glucose levels between 100 and 125 mg/dL and without a previous diagnosis of diabetes. Participants were excluded from the study if they did not consume a diet with daily presence of grains or cereals derivatives, tubers or/and legumes, or if they presented any other health problem that the research staff considered contraindicated.

Nine subjects were included (Appendix A Appendix A) and all participants provided written informed consent, and the study protocol was approved by the Ethics Committee of Hospital Clinic de Barcelona. Measure protocol can be observed in Appendix A.

### 2.2. Study Design

The study was a cross-over pilot clinical study consisting of two periods. The first period was only an observational and monitoring phase where participants maintained their regular diet (RD); for this reason, all participants were initiated in this period and a wash-out term was not needed. Subsequently, with the data of the first phase obtained, the subjects began the second period in which they had to undergo a nutritional intervention with a quinoa diet (QD).

A total of six visits plus two quinoa product collection days were programmed (Figure 1). After the pre-study visit (V0), which took place a week before the start of the nutritional intervention and where researchers obtained signed informed consent, participants were summoned for a first visit (V1) where they were explained how they should fill in the dietary records and they were applied with the FreeStyle Libre^®^.

Subjects then began RD, a period of 4 weeks during which only their normal life was monitored. After the first 14 days of this, a second visit (V2) was made where the dietary record was collected that would serve to account for their usual consumption of cereals, flours, tubers and legumes, the FreeStyle Libre^®^ sensor was also collected.

On the last day of RD period, day 28, they were cited (V3) in consultation where blood samples were taken after an 8 h fast, anthropometric measurements and blood pressure measurements were obtained, and the new FreeStyle Libre^®^ sensor was put in place. Participants were asked about their physical activity and exercise practice during those past 4 weeks by a short questionnaire adapted from the Minnesota Leisure Time Physical Activity Questionnaire for individuals of advanced age (VREM questionnaire), and a new empty 14-day dietary record was given. In addition, the volunteers received the first foods with quinoa to initiate QD the next day. Products were delivered weekly for conservation reasons but also to ensure that they followed an adequate consumption; they had to go through consultation to pick up the product and gave the researchers the empty packs where quinoa products had been.

On the next visit, day 42 (V4), the Freestyle Libre^®^ sensor was collected and the filled dietary record was collected. Finally, after 28 days of quinoa diet, they were summoned for the last visit, day 56 (V5), where all the determinations were repeated identically as V3.

### 2.3. Study Food

With the premise that the products created replaced not only grains, legumes and tubers, but also farinaceous commonly consumed by the participants and that only the cereal fraction was modified, similar products based on quinoa flour were created. The creation of these products was necessary, after conducting a market search where it was observed that there was not enough food to replace those consumed since these had percentages of quinoa flour not exceeding 20–30%.

Thus, apart from delivering quinoa, quinoa flakes and quinoa flour to the participants, they were given products created with ≥70% quinoa flour, which were biscuits, crackers, brioche, sponge cake, baguette bread, sliced bread and pasta (Appendix A) created and produced by Alicia-elBulli Foundation. Moreover, quinoa-based recipes were delivered with eight commonly consumed recipes that replaced the tubers, legumes or grains of the recipe. Each subject received the equivalent of what they consumed according to their RD dietary records. Thus, only if the volunteer had indicated that he consumed sponge cake was the quinoa-based product delivered to him.

### 2.4. Calculations and Statistical Analyses

Descriptive data are presented as the mean and standard deviation (SD) or median and interquartile range (IQR) for continuous variables, and the frequencies and percentages (%) for categorical variables. Anthropometric measurements, blood test variables and dietary intake were compared at different times using the non-parametric Wilcoxon signed-rank test because normality and equality of variance could not be assumed due to small sample size (*n* = 9). In order to compare variables related to dietary patterns, mean value for dietary intake, including all meals, was considered for each participant.

The glucose level monitoring sensor takes measurements at discrete time points for each patient (Appendix A). Therefore, firstly the glucose curves are linearly interpolated in order to have observations for each patient at equal time points (Appendix A). A first glance of the glucose curves over the day showed that they were more homogeneous around breakfast than around other later meal intakes. Therefore, the glucose concentration values corresponding to the breakfast were considered as a function of time in minutes over the interval *t* = [−30, 120], which begins half an hour before the start of breakfast and ends two hours later. Before constructing a functional model, the functional data were time aligned in order to reduce the differences between different patients and/or different days (for instance, some patients could mark the starting time of breakfast systematically better than others, or spend systematically more time at breakfast than the average) (Appendix A). The time alignment was performed by warping functions, using the function WFDA in the R package fdapace [16].

Once the glucose level curves were synchronized, a functional regression analysis was conducted to model the effect of diet type, patient and nutrient intake on monitored glucose levels. Three different explanatory variables were considered: diet type with two categories (regular and Quinoa diets), patient indicator (categorical variable with nine levels) and the contents of different nutrients. The breakfast glucose curves are handled as the functional response variable. To study the relationship between these variables, function on scalar regression (fosr) models were used. The functional regression model protocol can be observed in the Appendix A, which was fitted by penalized flexible functional regression, as implemented in the function pffr of the R package refund [17].

### 2.5. Data Availability Statement

The datasets generated and analyzed during the current study are not publicly available due to the authors exploiting the results in other analyzes and publishing the following results. However, datasets are available from the corresponding author upon reasonable request.

## 3. Results

### 3.1. Participant Characteristics

The baseline characteristics of nine patients participating in the study are detailed in Appendix A. The mean age was 69.6 (SD 2.8) years and 6 (66.6%) were female. The mean body mass index was 28.4 (SD 3.2) kg/m^2^, systolic blood pressure 130.4 (SD 14.6) mmHg and diastolic blood pressure 79.6 (SD 11.9) mmHg. Regarding comorbidities and risk factors, six out of nine (66.6%) patients had hypertension, four (44.4%) patients had hyperlipidemia and three (33.3%) patients had a family history of diabetes mellitus.

### 3.2. Anthropometric Measurements and Blood Tests

Anthropometric measurements and blood test variables for the study participants are shown in Table 1. To evaluate changes pre and post QD, we analyzed the changes produced between V3 and V5, having previously verified that there were no changes in the physical activity performed by the participants during the RD and QD periods (data not shown). During the study period, patients’ weight (74.8 kg on the V3 vs. 73.2 kg on the V5, *p* = 0.008), BMI (26.8 kg/m^2^ on the V3 vs. 26.2 kg/m^2^ on the V5, *p* = 0.004) and waist circumference (93.5 cm on the V3 vs. 92.0 cm on the V5, *p* = 0.015) were slightly reduced. On the other hand, glucose levels had previously decreased (102 mg/dL on the V1 vs. 97 mg/dL on the V3, *p* = 0.021) after four weeks with the regular diet, and maintained during the time the quinoa diet lasted (97 mg/dL on the V3 vs. 96 mg/dL on the V5, *p* = 1.000). C-reactive protein values decreased; although, this decrease was not strong enough to be significant (7 mg/L on the V3 vs. 6.9 mg/L on the V3, *p* = 0.058). Levels of glycated hemoglobin (HbA1c) were also reduced during the study period (6.1% on the V1 vs. 5.5% on the V5, *p* = 0.007), which was not measured in V3 due to the short period of time between V1 and V3 or V3 and V5, because the half-life of erythrocytes, where HbA1c is detected, is at least 2–3 months [18].

### 3.3. Dietary Intake Differences

Nutrition patterns between the regular and the quinoa diet were compared by means of dietary intake differences. From 160 parameters related to the dietary patterns, 37 (23.1%) showed statistically significant differences between the regular and the quinoa diet periods (Appendix A). A slightly higher carbohydrate content was observed in the regular diet compared with the quinoa diet (32.1 g/100 g for RD vs. 28.3 g/100 g for QD, *p* = 0.004). The saturated fat content during the period corresponding to the quinoa diet was higher than the period corresponding to the regular diet (4.0 g/100 g for RD vs. 5.1 g/100 g for QD, *p* = 0.012). The content of lipids was also higher in the quinoa diet (15.3 g for RD vs. 19.9 g for QD, *p* = 0.004). Regarding the amino acid profile, a higher content of cystine (170.6 mg for RD vs. 146.3 mg for QD, *p* = 0.008), arginine (699.7 mg for RD vs. 591.7 mg for QD, *p* = 0.008), glutamic acid (2339 mg for RD vs. 1981.4 mg for QD, *p* = 0.008) and proline (755.7 mg for RD vs. 667.7 mg for QD, *p* = 0.039) was related to the quinoa diet compared with the regular diet.

### 3.4. Results of Functional Regression Models

The functional regression model using only the diet factor was represented in Figure 2. This simplest model, which has an adjusted R-squared value of 0.343, evidences the positive effect of the quinoa diet in reducing glucose concentrations. Globally, the effect of the quinoa diet starts right after finishing breakfast (around minute 30) and lasts until one hour and a half after breakfast. The functional regression model with two factors using both the diet and the patient factor slightly increased the goodness of fit (adjusted R-squared is 0.404). In this case (Figure 3), different patients have different patterns because, by construction, the nine individual curves have a sum equal to the constant function zero. The previous model was extended to include scalar nutritional parameters in the model. Of the 160 variables related to the nutrition patterns, 18 were discarded for lack of variance (variance equal to 0) and 10 were discarded to avoid multicollinearity (a correlation coefficient equal to or higher than 0.99 with other variables). A functional regression model using the diet factor, the patient factor and 132 nutritional variables was fitted. Then, this model was simplified by removing the non-significant functional coefficients. According to this, 43 nutritional variables were removed from the model. From the 89 remaining nutritional variables, the coefficients of 40 were considered as almost constant functions (equivalent degrees of freedom lower than 1.1, while this value is exactly 1 for a constant function). The final model, including the diet factor, the patient factor, 40 nutritional variables treated as constant coefficients, and 49 nutritional variables treated as functional coefficients, had an adjusted R-squared equal to 0.734. The significance of the constant and the functional coefficients is detailed in Figure 4. Five constant coefficients and one functional coefficient were considered as non-significant for the model. In this model, higher contents of vegetable fiber, adrenine, MUFA, carbohydrates and lipids were related to higher glucose concentrations, while higher contents of energy, guanine, TRAP (evaluation of Total Reactive Antioxidant Potential), isoleucine and vitamin K were associated with lower glucose concentrations (Figure 5). Regarding the functional-coefficient patterns of nutritional variables (Figure 6 shows the 12 most significant ones), the most significant with augmenting effects are γ-tocopherol, soluble fiber and ORAC (Oxygen Radical Absorbance Capacity), and those with the most significant reducing effects are theobromine, FA, fructose, phytic acid, PUFA w6, citric acid, cellulose and % of energy from proteins. The effect of non-soluble fiber is also highly significant, as it increased glucose levels at the end of the studied period.

## 4. Discussion

Dietary pattern changes have demonstrated an important link in preventing several chronic diseases through multiple associated ways. In particular, the prevention of T2D due to changes in diet, more specifically, studies based on the supplementation of specific food or the substitution of one food for another, have been extensively investigated.

Despite the large increase in the publications of these types of studies and the important consumption of quinoa as a trend, the benefits of its consumption have never been studied, in substitution of other grains commonly consumed, in the prevention of T2D.

In line with results we have found, the substitution of grains and flours from other sources as well as tubers and legumes, without calorie restriction, has shown an improvement in the management of body weight, BMI and waist circumference, despite the fact that no previous study with quinoa suggested anthropometric changes [1,14]. It is probably because, although no differences in energy consumption were observed, there was a change in the nutritional pattern of individuals. We observed a lower consumption of carbohydrates, polysaccharides and starch, and a higher consumption of fats due to the products delivered because quinoa contains fewer carbohydrates and more fats [1]. Specifically, the differences in nutrients observed in the quinoa diet phase are attributable to the change in the pseudocereal and not to other dietary modifications since quinoa is rich, among other things, in unsaturated fats and protein, and poor in carbohydrates. In this line, the thermodynamic arguments imply that the number of calories is not the only determinant of body weight, but that the composition of the diet also plays a very important role [19]. Precisely, the difference in weight that we can observe is due to these three macronutrients. High carbohydrate diets increase insulin secretion, which consequently promotes fat storage in the adipose tissue and induces a decrease in metabolic rate resulting in further fat mass storage [20]. On the other hand, since dietary fats do not stimulate insulin secretion, high fat isocaloric diets reduce insulin secretion promoting fat loss from adipose tissue, turning free-FA available for use by metabolically active tissues [21]. Finally, dietary proteins are known to positively influence fat-free mass during weight loss [22]; thus, high protein diets offer benefits in terms of energy expenditure and body composition.

Despite the differences in weight between the two groups in our study, it has been shown that the quality of the diet is more important in a group of older people with preDM than body weight [23]. Therefore, quinoa’s low glycemic index, compared to other commonly consumed cereals, may be responsible for improvements in blood glucose fluctuations, as previously seen [24]. The problem we observed with venous glucose measuring was that, during the first 4 weeks of monitoring in their regular diet, the participants were able to visualize the fluctuations in their blood glucose levels because they were using their FreeStyle Libre^®^ sensor. It has been observed that, with the knowledge of blood glucose in real time, the participants make decisions about their meals to improve them. Therefore, after decreasing venous basal glucose during a regular diet, the quinoa diet was probably not sufficient for 4 weeks to show a decrease in that previously decreased basal glucose.

The functional regression model allowed us to observe that the consumption of a diet rich in quinoa has a drafting effect on blood glucose fluctuations compared to the usual diet, as previously observed by comparing the consumption of wheat and that of quinoa [14,25]. We can also detect that the behavior of these fluctuations in the diet rich in quinoa is different depending on each participant; as already observed in a study with a cohort of 800 people, there is an interpersonal variability of glycemic response [26].

It is well-known that the amount of carbohydrates present in a meal can affect postprandial blood glucose, but it has also been indicated that only counting carbohydrates cannot be definitive in predicting the blood glucose level after mixed meals [27]. Therefore, the presence of fiber, protein or fat can decrease this postprandial blood glucose peak [28] and quinoa is a food that not only has fewer carbohydrates but also has significant amounts of fiber, protein and healthy fats. Many other nutrients, despite being present in smaller amounts, can influence the postprandial response [29], even a group of nutrients together creating a synergy [30,31].

With more precise analysis, we observed that other nutritional variables played an important role in modulating glucose fluctuations in our subjects.

We observed that the more energy, the lower the glucose values. This was probably due to the fact that, despite maintaining a similar energy intake between products consumed in the RD and the QD, quinoa has diverse nutrients that, as we have said, may imply lower glycemia [1]. Likewise, antioxidant molecules and protein nutrients are also related to the same glycemic response in the QD, as previously has been observed [6,32,33]. Conversely, carbohydrates, as we could expect, but also fats, especially monounsaturated fatty acids (MUFA), were associated with higher glucose concentrations in this diet.

When we look at the more specific nutrients with a greater significance, we observe that the nutrients that promote the decrease in glycemia are those that are present in quinoa: cis FA, PUFAw6, cellulose and phytic acid [34,35,36]. On the contrary, nutrients that promote and increase glucose with the greatest significance are those not highly present in quinoa, but rather in other foods, and these are soluble fiber [37] and γ-tocopherol [34,38]. Thus, we can be assured that the benefits that we observed in the fluctuations of glycemia in the quinoa diet were due to the quinoa itself and not due to the modification of other foods or nutrients not present in the quinoa.

Because the preDM of our participants was measured by fasting glucose, they therefore presented with impaired fasting glucose and OGTT was not measured, so we do not know if they also presented with impaired glucose tolerance. In many cases, the two situations overlap to produce this state prior to the development of T2D where insulin resistance is in both muscle and liver [38]. All the volunteers in our study were overweight or obese, so weight loss is a good prognosis for delaying the appearance of T2D [11], but the quality of the diet in a group of elderly and preDM people may be more interesting [23].

Postprandial glycemia is an independent risk factor for the development of T2D [9], and it has been demonstrated that the decrease in postprandial glycemia could decrease the toxic effects of glucose and delay the conversion of impaired glucose tolerance to T2D [39]. Thus, it is necessary to make food choices that induce a normal postprandial glycemic response [40]. In this sense, in our study, we have shown that quinoa, thanks to its beneficial effect on postprandial glycemia, could be a good dietary choice to include in the regular consumption of the population with preDM.

## 5. Strengths

The fact of doing a cross-over-designed study and not standardizing the quantities of quinoa and quinoa products allows us to observe how their consumption in substitution of other foods (cereals, tubers and legumes) affects the real life of the participants.

Other studies carried out have obtained variable results among them, probably due to the different objectives, which led to unlike methodologies and measurements, divided doses and differences in the food delivered compared to the standard diets [14,41,42,43,44].

## 6. Limitations

The small number of subjects studied, because we carried out a pilot study, makes it difficult to obtain more robust results, but for this type of super personalized and controlled intervention, the sample size is deemed appropriate by most researchers. On the other hand, the duration of our study represents a short period in the context of lifelong dietary exposures. In this context, the effects of a longer intervention, in a larger sample size, would be interesting to observe.

## 7. Conclusions

In a population of advanced age and at risk of developing T2D subjects, a diet rich in quinoa as a substitution for other foods rich in complex carbohydrates commonly consumed is very promising as a T2D-preventive strategy. A diet rich in quinoa reduces postprandial glycemia despite intrapersonal differences, thanks to the joint action of different nutrients and the suppression of others consumed in a regular diet, which could apply a brake to the progression to T2D.

## Figures and Tables

**Figure 1 nutrients-14-02331-f001:**
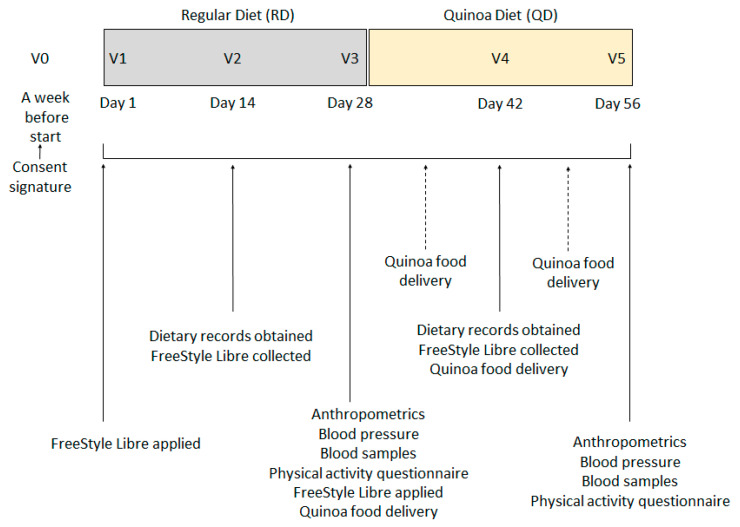
Study design and visits programmed.

**Figure 2 nutrients-14-02331-f002:**
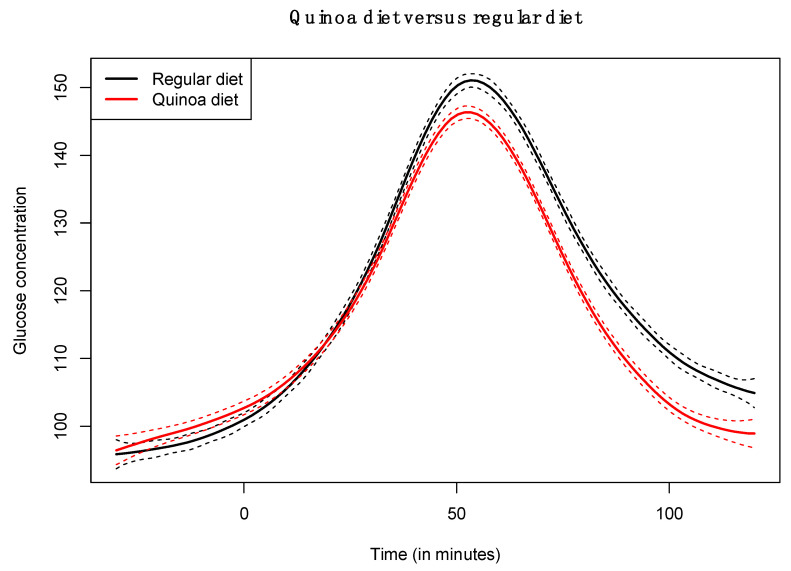
Coefficient estimate of functional regression model with diet factor.

**Figure 3 nutrients-14-02331-f003:**
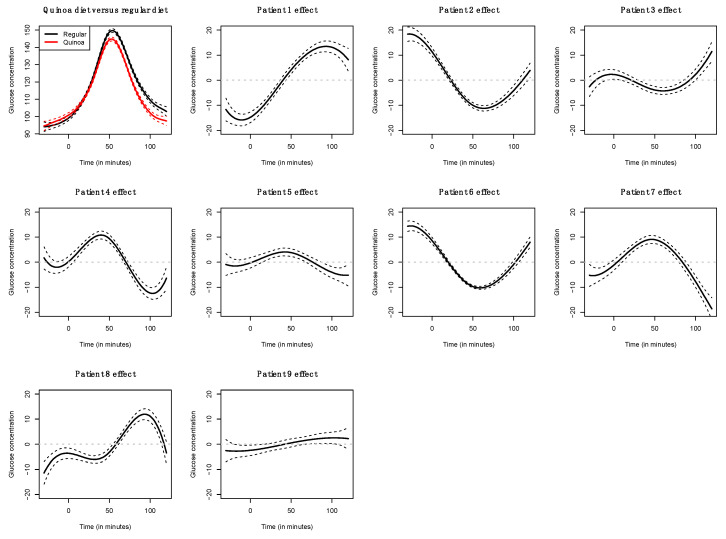
Coefficient estimates for the functional linear model with diet factor and patient effect.

**Figure 4 nutrients-14-02331-f004:**
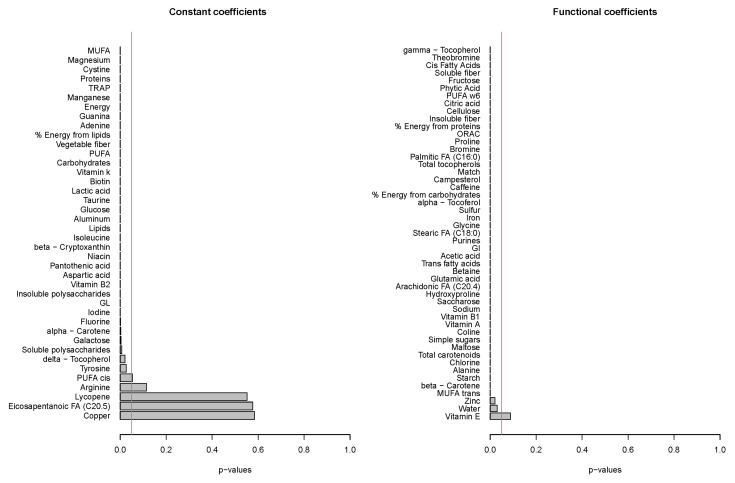
The significance of the constant and functional model coefficients.

**Figure 5 nutrients-14-02331-f005:**
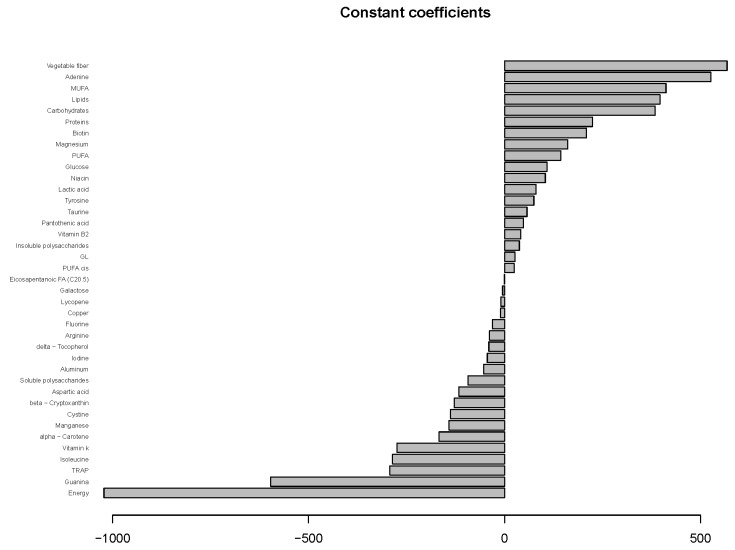
Constant coefficient estimates of the final model.

**Figure 6 nutrients-14-02331-f006:**
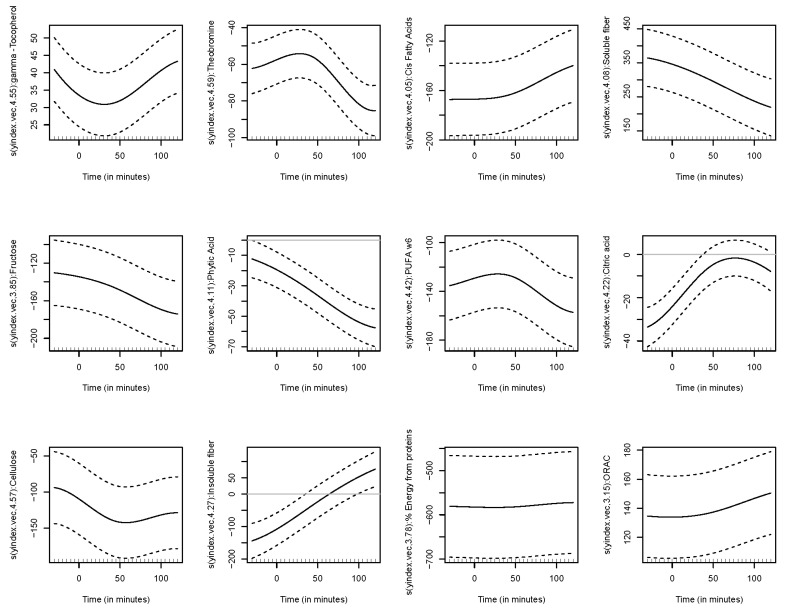
Functional coefficient patterns of nutritional variables.

**Table 1 nutrients-14-02331-t001:** Anthropometric measurements and blood test variables from 9 participants.

Variables	V1	V3	V5	*p*-Value		
	Median (Q1, Q3)	Median (Q1, Q3)	Median (Q1, Q3)	V1 vs. V3	V3 vs. V5	V1 vs. V5
Weight	75 (69, 76.5)	74.8 (69, 77.4)	73.2 (68, 76.7)	0.075	**0.008**	**0.021**
BMI	26.6 (26.2, 31.2)	26.8 (26.5, 31.1)	26.2 (26.1, 30.4)	0.063	**0.004**	**0.021**
Waist	93 (86.5, 96.5)	93.5 (84, 96)	92 (83, 95.7)	0.310	**0.015**	**0.012**
Hip	104.9 (100, 109)	105.5 (100, 110)	103 (100, 105.5)	0.722	**0.043**	0.213
Waist/hip	0.89 (0.83, 0.92)	0.86 (0.84, 0.91)	0.87 (0.82, 0.89)	0.398	0.281	0.182
SBP	128.7 (124.7, 135)	126.3 (121.3, 136)	125.7 (122.2, 133.2)	0.859	0.446	0.109
DBP	84.7 (74.7, 85.3)	78.3 (76, 89.3)	75.7 (71.3, 88.8)	0.678	0.055	0.833
HR	66 (62.7, 67)	66.3 (62.3, 69)	66.8 (63, 71.2)	0.496	0.641	0.293
Fasting glucose	102 (100, 114)	97 (96, 101)	96 (93, 100)	**0.021**	1.000	**0.021**
Hs-CRP	0.35 (0.23, 0.72)	0.23 (0.13, 0.36)	0.21 (0.12, 0.4)	1.000	0.734	0.917
CRP	6.8 (6.5, 6.9)	7.0 (6.9, 7.2)	6.9 (6.7, 7.0)	**0.027**	0.058	0.223
Cholesterol	182 (163.5, 194)	195 (179, 220)	182 (174, 198)	0.156	0.263	0.446
Triglycerides	71 (56.5, 113.5)	90 (80, 121)	86 (69, 132)	0.463	0.477	0.686
HDL	49 (44.5, 58)	52 (47, 63)	51 (50, 54)	1.000	0.726	0.799
LDL	110 (103.5, 142)	127.6 (97, 141.8)	108 (104.4, 137.2)	0.469	0.250	0.938
Serum albumin		43 (42, 46)	43 (41, 45)		0.565	
Insulin		12.3 (10.7, 16.8)	13 (9.1, 18.2)		1.000	
HbA1c	6.1 (5.9, 6.3)		5.5 (5.3, 5.6)			**0.007**

BMI, body mass index; SBP, systolic blood pressure; DBP, diastolic blood pressure; HR, heart rate; Hs-CRP, high-sensitivity C-reactive protein; HDL, high-density lipoprotein; LDL, low-density lipoprotein. Values are presented in median and quartiles. In bold, *p* ≤ 0.05 using the Wilcoxon signed-rank test.

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
