# Peer review of "Glycaemia Fluctuations Improvement in Old-Age Prediabetic Subjects Consuming a Quinoa-Based Diet: A Pilot Study"

_nutrients, 2022, doi:10.3390/nu14112331_

Round 1

Reviewer 1 Report

I have read with high interest this paper. I want to congratulate the authors, for their efforts and interest in this field, often neglected. However, I have several considerations:

  1. Extensive grammar review is needed, in both main text and annexes
  2. The number of enrolled patients and the follow-up is too low to conclude that quinoa-based diets might have an effect in preventing diabetes development. The improvement in HbA1c and other parameters can be attributed to other confounding factors; thus, a multivariate regression analysis is needed. The unblinded design also contributes to a possible Hawthorne effect.
  3. The definition of diabetes is incomplete, as HbA1c is not considered (Which is a diagnostic criterion according to the American Diabetes Association), and if you follow the WHO criteria, 2-hour oral glucose tolerance test is needed.
  4. The authors do not provide basic information about continuous interstitial glucose monitoring: Baseline glucose levels, peak glucose after meal, glucose variability… And if they used only the main meal or also included other meals (dinner, breakfast…) in their analysis. If they mixed all meals, this could be considered as a bias (Supper is normally more frugal, in comparison with breakfast or lunch, and glucose response is different, as well as insulin resistance). The authors should analyze meals separately and provide disaggregated data.
  5. Exercise patterns should be also analyzed, as changes in physical activity can also affect insulin sensitivity and blood glucose
  6. Patients 5 and 7 CAN BE CONSIDERED AS PATIENTS WITH DIABETES, as they have baseline HbA1c levels of 6.5 and above, following the American Diabetes Association criteria. This is a serious bias.
  7. There are, as you know, several drugs that impair glucose metabolism as well as gastric motility. Please provide information about use of pharmacological compounds before starting the trial and between days 28 and 52
  8. Prediabetes can be also diagnosed if HbA1c is between 5.7 and 6.5%, even with FPG below 100 mg/dl. Why did you exclude patients with fasting glucose <100 mg/dl, without checking HbA1c?
  9. HOMA-IR should be calculated and displayed
  10. Patient No. 3 had a cardiovascular event. Please elaborate further.
  11. Mediterranean diet is known for improving HbA1c. Did you use any Mediterranean diet adherence questionnaire? In you did, please include the results. If not, it could be considered as an important bias.
  12. As you know, the glucose-induced secretion of insulin may be as much as doubled in the presence of amino acids, such as arginine and lysine. You analyzed arginine, which is increased in the RD:
  • Why did you not analyze lysine?
  • Could this difference between diets, in terms of amino-acid composition, have an effect in HbA1c final values?
  1. In Suppl. Table 3, please include measure units.
  2. In Suppl. Figure 2, yellow should be avoided (Visually inconspicuous). Also, some patients have glucose levels of >200 mg/dl… What were the baseline HbA1c values of these patients?
  3. While I do not see any normality tests (Shapiro-Wilks would be advisable), I think that it is highly possible that your population follows a non-parametric distribution. I suggest avoiding mean and SD in such a scarce population, and instead using median and interquartile range, to avoid the effect of outliers.

Author Response

All the authors would like to jointly thank you for your extensive review and all your proposals. Next we proceed to expose the changes made and some answers to your questions

I have read with high interest this paper. I want to congratulate the authors, for their efforts and interest in this field, often neglected. However, I have several considerations:

  1. Extensive grammar review is needed, in both main text and annexes

A grammar review has been made in manuscript but also in annexes too. 

  1. The number of enrolled patients and the follow-up is too low to conclude that quinoa-based diets might have an effect in preventing diabetes development. The improvement in HbA1c and other parameters can be attributed to other confounding factors; thus, a multivariate regression analysis is needed. The unblinded design also contributes to a possible Hawthorne effect.

We assume that your comment is related to the variables listed in Table 1, and that you are suggesting fitting a regression model with multivariate response (being the differences between V1 and V5 for all the 18 variables in Table 1) and 132 nutritional explanatory variables. Due to the limited sample size (n=9), this multivariate regression model can not be fitted by OLS. Even a one-dimensional response linear model can not be fitted because the number of observations (n=9) is lower than the number of variables (p=132). On the contrary, it is possible to fit a functional linear regression model to analyze the effect of nutrients and the diet type on glucose levels, because it uses many observations over time for each patient.

A blinded design turned out to be impossible because, for this pilot study, it was necessary to know the usual consumption of the food groups in order to then calculate them and deliver the quinoa-based products. We know that unblinded studies are more limited, but due to the type of pilot study, it was necessary to do so.

  1. The definition of diabetes is incomplete, as HbA1c is not considered (Which is a diagnostic criterion according to the American Diabetes Association), and if you follow the WHO criteria, 2-hour oral glucose tolerance test is needed.

We did not define type 2 diabetes, we only limit ourselves to describe how participants with prediabetes were detected “Subjects included, [...] were males and females ≥65 years old with fasted glucose levels between 100 and 125 mg/dL and without a previous diagnosis of diabetes. ”. This is because in Health Public System in Spain, HbA1c is not usually obtained from people who has not previuosly presented >126mg/dL of fasted glucose so looking at HbA1c was not an option. Moreover, HbA1c could not be a main outcome in our project due to the change time it requires as we already explained in “Levels of glycated hemoglobin (HbA1c) [...] which was not measured in V3 due to the short period of time between V1 and V3 or V3 and V5 due to half-life of erythrocytes, where HbA1c is detected, is at least 2-3 months [18]”. 

  1. The authors do not provide basic information about continuous interstitial glucose monitoring: Baseline glucose levels, peak glucose after meal, glucose variability… And if they used only the main meal or also included other meals (dinner, breakfast…) in their analysis. If they mixed all meals, this could be considered as a bias (Supper is normally more frugal, in comparison with breakfast or lunch, and glucose response is different, as well as insulin resistance). The authors should analyze meals separately and provide disaggregated data.

The reason why function on scalar regression (fosr) models was used and different formulas was created is because we calculate peaks after meal considering baseline glucose levels of each participant and glucose variability from each volunteer. More information regarding this could be observed in Supplemental Figure 2 but also in  Supplementary information file where formulas were explained. In relation with meals, we used only breakfast as the reviewer can observe in this methods sentence: “the glucose concentration values corresponding to the breakfast were considered as a function of time in minutes over the interval t= [-30,120], that begins half an hour before the start of breakfast and ends two hours later”. The reason for choosing breakfast to observe the variations in postprandial glucose fluctuations was that it was the only time of the day that we could ensure a sufficient fast for all the participants that would not condition the following meal.

  1. Exercise patterns should be also analyzed, as changes in physical activity can also affect insulin sensitivity and blood glucose

Exercise patterns were also controlled as explained in methodology “Participants were asked about their physical activity and exercise practice during those past 4 weeks by a short questionnaire adapted from the Minnesota Leisure Time Physical Activity Questionnaire for individuals of advanced age (VREM questionnaire)” and in Figure 1 you cna observe Physical activity questionnaire in both V3 and V5. But, we consider after your comments that this information not clearly understandable. For this reason, we decided to include this argument “To evaluate changes pre and post QD, we analyze the changes produced between V3 and V5, having previously verified that there were no changes in the physical activity performed by the participants during the RD and QD periods [data not shown] ” in Results part. 

  1. Patients 5 and 7 CAN BE CONSIDERED AS PATIENTS WITH DIABETES, as they have baseline HbA1c levels of 6.5 and above, following the American Diabetes Association criteria. This is a serious bias.

As the ADA has published in its guidelines this January 2022, the HbA1c value alone does not confirm the diagnosis of T2D if blood glucose is not above 126. In the absence of unequivocal hyperglycemia, diagnosis requires two abnormal test results from the same sample or in two separate samples”. 

American Diabetes Association; Standards of Medical Care in Diabetes—2022 Abridged for Primary Care Providers. Clin Diabetes 1 January 2022; 40 (1): 10–38. https://diabetesjournals.org/clinical/article/40/1/10/139035/Standards-of-Medical-Care-in-Diabetes-2022

  1. There are, as you know, several drugs that impair glucose metabolism as well as gastric motility. Please provide information about use of pharmacological compounds before starting the trial and between days 28 and 52

In the eligibility criteria, it was determined that the use of certain drugs was a reason for exclusion from the study, as well as having suffered certain pathologies or presenting them actively. This information can be found available to everyone in the official study registry: https://clinicaltrials.gov/ct2/show/NCT04529317

The clinical researchers checked that in no case was there a change in the medication of the patients during the performance of the nutritional intervention.

  1. Prediabetes can be also diagnosed if HbA1c is between 5.7 and 6.5%, even with FPG below 100 mg/dl. Why did you exclude patients with fasting glucose <100 mg/dl, without checking HbA1c?

This answer can be found in the respond of question 3. 

  1. HOMA-IR should be calculated and displayed

We decided to base our results on the postprandial blood glucose fluctuations that we found because it was the most interesting part of the study thanks to the use of continuous glucose monitors and the capacity that we hypothesized that the substitution of quinoa grains would have. We do not believe that the calculation of HOMA-IR is interesting because, as we explained, the patients decreased their basal glucose in the RD “The problem we observed with venous glucose measure was that, during the first 4 weeks of monitoring in their regular diet, the participants were able to visualize the fluctuations in their blood glucose levels because they were using their FreeStyle Libre® sensor. It has been observed that, with the knowledge of blood glucose in real time, the participants make decisions about their meals to improve them. Therefore, after decreasing venous basal glucose during a regular diet, the quinoa diet was probably not sufficient for 4 weeks to show a decrease in that previously decreased basal glucose”.

  1. Patient No. 3 had a cardiovascular event. Please elaborate further.

Participant number 3 had a cardiovascular event years before inclusion in the study. Therefore, he did not meet any of the exclusion requirements described and was eligible to enter the study https://clinicaltrials.gov/ct2/show/NCT04529317.

  1. Mediterranean diet is known for improving HbA1c. Did you use any Mediterranean diet adherence questionnaire? In you did, please include the results. If not, it could be considered as an important bias.

Thank you for this observation about the Mediterranean diet since, due to our residence in Barcelona, it is something that the population has very internalized. On the other hand, Mediterranean diet is a T2D preventive type diet but is not the only one. Specifically, and as the ADA explains (*), there is no single type of beneficial diet, so we consider it more important, following the line of clinical guidelines, to identify healthy foods and nutrients in dietary patterns. For this reason, we decided to analyze the nutritional patterns specifying the consumption of each one of them instead of obtaining only a score questionnaire of approach to the Mediterranean diet.

(*) American Diabetes Association Professional Practice Committee; 3. Prevention or Delay of Type 2 Diabetes and Associated Comorbidities: Standards of Medical Care in Diabetes—2022. Diabetes Care 1 January 2022; 45 (Supplement_1): S39–S45. https://doi.org/10.2337/dc22-S003

  1. As you know, the glucose-induced secretion of insulin may be as much as doubled in the presence of amino acids, such as arginine and lysine. You analyzed arginine, which is increased in the RD:
  • Why did you not analyze lysine?
  • Could this difference between diets, in terms of amino-acid composition, have an effect in HbA1c final values?

160 parameters related to the dietary patterns were analysed by DIAL program, 37 (23.1%) shown statistically significant differences between the regular and the quinoa diet periods and were the only included in Supplementary Table 3. Lysine was also analysed and results were: RD 4081,43mg (SD 726,06) and QD 4084,08mg (SD 426,73) with p-value 0.99 but we are unable to show the 160 nutrients in this article. 

From the all amino-acids studied (Cistina, Histidina, Isoleucina, Leucina, Lisina, Metionina, Fenilalanina, Serina, Treonina, Triptófano, Tirosina, Valina, Arginina, Ácido Glutámico, Alanina, Ácido Aspártica, Glicina, Prolina, Hidroxiprolina, Taurina ) only arginine and glutamic acid were the only with statistically significant differences. 

  1. In Suppl. Table 3, please include measure units.

The units of measurements have been included accordingly. 

  1. In Suppl. Figure 2, yellow should be avoided (Visually inconspicuous). Also, some patients have glucose levels of >200 mg/dl… What were the baseline HbA1c values of these patients?

We have taken the reviewer’s advice and avoided yellow lines. Supplementary figure 2 represents raw, interpolated, and aligned data of patient 1 during regular diet period. Thus, each line corresponds glucose concentrations of patient 1 in different days. Baseline HbA1c of this patient was 5.8%. 

  1. While I do not see any normality tests (Shapiro-Wilks would be advisable), I think that it is highly possible that your population follows a non-parametric distribution. I suggest avoiding mean and SD in such a scarce population, and instead using median and interquartile range, to avoid the effect of outliers.

This is a good suggestion. We used median and interquartile range in table 1 and supplementary table 1 as has been suggested by the reviewer.  

Very thankful, 

All the authors.

Reviewer 2 Report

Regarding the manuscript "Glycaemia fluctuations improvement in old-age prediabetic subjects consuming a quinoa-based diet: a pilot study"the following issues could be mentioned:

  1. What is the term "fosr model" in the abstract?
  2. The references for the relation between type 2 diabetes and quinoa consumption used in the Introduction are not that relevant for the study objective. Literature research could bring more references.  
  3. What is the reference for the first phrase in the Abstract? it is not clear if it can be found in the Introduction.
  4. Glucose sensors are useful for patients already diagnosed with diabetes, but are not that accurate to be used in clinical studies. Venous glycemia is the standard parameter to be used.
  5. The number of persons included is quite small.
  6. One patient had HbAc1 of 6.7% and met the criteria for type 2 diabetes. What was the fasting glycemia level?
  7. What are the references for the used questionnaires for assessing physical activity and food intake?
  8. All flours, tubers, legumes and farinaceous were replaced by quinoa?
  9. Where were the quinoa products created? 
  10. CRP is usually measured in mg/L
  11. What was the purpose of representing functional coefficient patterns of nutritional variables and not the dietary intake differences between regular and quinoa diets in the manuscript? 
  12. The conclusions of the study could be toned down since the short-term study has not proven that quinoa consumption decreases diabetes incidence.

Author Response

All the authors would like to jointly thank you for your extensive review and all your proposals. Next we proceed to expose the changes made and some answers to your questions

Regarding the manuscript "Glycaemia fluctuations improvement in old-age prediabetic subjects consuming a quinoa-based diet: a pilot study"the following issues could be mentioned:

  1. What is the term "fosr model" in the abstract?

Thank you for the remark. In the fourth sentence of the abstract we mentioned that the fosr is the abbreviation of “function on scalar regression”. 

  1. The references for the relation between type 2 diabetes and quinoa consumption used in the Introduction are not that relevant for the study objective. Literature research could bring more references.  

The scarce existing bibliography on quinoa and type 2 diabetes did not allow a great extension of the references. In fact, in the cases of reference 1 and 13, they are cited reviews that include different studies. We have observed that in February of this year a new study* has been published that we have already included in our article. 

(*) Zhang Y, Bai B, Yan Y, Liang J, Guan X. Bound Polyphenols from Red Quinoa Prevailed over Free Polyphenols in Reducing Postprandial Blood Glucose Rises by Inhibiting α-Glucosidase Activity and Starch Digestion. Nutrients. 2022 Feb 9;14(4):728. doi: 10.3390/nu14040728. PMID: 35215378; PMCID: PMC8875175.

  1. What is the reference for the first phrase in the Abstract? it is not clear if it can be found in the Introduction.

The references quotes from 1 to 7 are the abstract that makes us think of the first sentence of the abstract which are explained in the first paragraph of introduction. 

“Quinoa is a pseudo-cereal and has potential health benefits and exceptional nutritional value: a high concentration of protein with all essential amino acids, unsaturated fatty acids (FA), vitamins and minerals [1]. Furthermore, quinoa is rich in fiber and complex carbohydrates both making it a low glycemic index grain [1]. Among other nutrients, phenolic compounds founded in quinoa [2] [3] show inhibitory effects on α-glucosidase and lipase activities [4] [5] which are involved in sugar and lipid digestion in the digestive tract, respectively. Specifically, it has recently been observed that the polyphenols present in quinoa could have an effect on the reduction of postprandial blood glucose [6]. All this leads us to think that quinoa can be used to improve the profile of metabolic risk factors and help control type 2 diabetes (T2D) [3] [4] [7]”.

  1. Glucose sensors are useful for patients already diagnosed with diabetes, but are not that accurate to be used in clinical studies. Venous glycemia is the standard parameter to be used.

It is not possible to use venous glucose to observe postprandial glucose changes in daily consumption of a food for weeks for the study of effects in real-time in real-life. In addition, the use of continuous monitoring sensors is increasingly used not only by the population but also by scientific studies.

  1. The number of persons included is quite small.

To model the glucose curves as a function of time, the number of discrete time points where the sensor takes measurements is quite important. Although the sample size was small, the sensor was taking measurements every 15 minutes, so we had enough number of discrete points to apply functional models.  

  1. One patient had HbAc1 of 6.7% and met the criteria for type 2 diabetes. What was the fasting glycemia level?

All patients presented fasted glucose levels between 100 and 125mg/dL. Specifically, he had a fasting glycemia of 114 mg/dL. As the ADA has published in its guidelines* this January 2022, the HbA1c value alone does not confirm the diagnosis of T2D if blood glucose is not above 126. In the absence of unequivocal hyperglycemia, diagnosis requires two abnormal test results from the same sample or in two separate samples”. 

(*)American Diabetes Association; Standards of Medical Care in Diabetes—2022 Abridged for Primary Care Providers. Clin Diabetes 1 January 2022; 40 (1): 10–38. https://diabetesjournals.org/clinical/article/40/1/10/139035/Standards-of-Medical-Care-in-Diabetes-2022

  1. What are the references for the used questionnaires for assessing physical activity and food intake?

Food intake: in Supplementary information file this explanation can be found: “Nutritional patterns were measured using a 14-days dietary record revised corrected by a nutritionist and analyzed by the DIAL nutritional calculation program”. DIAL program was created by the Complutense University of Madrid and is a validated tool. 

Physical activity questionnaire: Ruiz Comellas A, Pera G, Baena Díez JM, Tudurí XM, Alzamora Sas T, Elosua R, et al. Validation of a Spanish Short Version of the Minnesota Leisure Time Physical Activity Questionnaire (VREM). Rev. Esp. Salud Publica vol.86 no.5 Madrid dic./oct. 2012

  1. All flours, tubers, legumes and farinaceous were replaced by quinoa?

Yes. Volunteers obtained different documents with the information to make the conversion in QD and the nutritionist explained them everything related to food substitution

  1. Where were the quinoa products created? 

All quinoa products were created in Fundación Alicia coordinated by Elena Roura (4th author). is a kitchen research center created by chef Ferran Adria.

https://www.alicia.cat/es/alicia/la-fundacion

  1. CRP is usually measured in mg/L

Hs-CRP (high sensitivity C reactive protein) was described in mg/dL because is the most common form measured. CRP is already changed. 

  1. What was the purpose of representing functional coefficient patterns of nutritional variables and not the dietary intake differences between regular and quinoa diets in the manuscript? 

Thank you for your comment. In this study, we considered the continuous structure of the glucose concentrations provided by each participant and handled them as a function of time. This way, we would be able to test the effect of the type of diet on the glucose levels measured in intended time period by using type of diet as independent variable and glucose functions as dependent variable. This allows us to analyze the effect of diet type on a continuous time interval.

  1. The conclusions of the study could be toned down since the short-term study has not proven that quinoa consumption decreases diabetes incidence.

Our conclusion is related to a T2D-prevetive stratgey “In a population of advanced age and at risk of developing T2D, a diet rich in quinoa as a substitution for other foods rich in complex carbohydrates commonly consumed, is very promising as a T2D-preventive strategy”. Afterthat, we only comment on the possibility that its use jointly with the supression of other complex carbohydrates products regulary consumed, could cause a decrease in the T2D incidence in a long term. 

Very thankful, 

All the authors.

Reviewer 3 Report

In the current study, promising results were observed with the Quinoa Diet, and the methodology is appropriate.

But, there are some questions. The authors reported “The problem we observed with venous glucose measure was that, during the first four weeks of monitoring in their regular diet, the participants were able to visualize the fluctuations in their blood glucose levels because they were using their FreeStyle Libre® sensor. It has been observed that, with the knowledge of blood glucose in real-time, the participants make decisions about their meals to improve them”. So, maybe the participants could also modify their diet during the Quinoa period. How could the authors validate that the participants did not change their diet?

The authors used the FreeStyle Libre® sensor for glucose measurements. Why did they not use the indices from the sensor (time in range, GMI, CV, and others) in their statistical analysis?

Furthermore, the authors did not report MARD (Mean absolute relative difference)   during both periods of their experiments.

Overall, the study could be interesting if the authors added the above data.

Author Response

All the authors would like to jointly thank you for your extensive review and all your proposals. Next we proceed to expose the changes made and some answers to your questions

In the current study, promising results were observed with the Quinoa Diet, and the methodology is appropriate.

But, there are some questions. The authors reported “The problem we observed with venous glucose measure was that, during the first four weeks of monitoring in their regular diet, the participants were able to visualize the fluctuations in their blood glucose levels because they were using their FreeStyle Libre® sensor. It has been observed that, with the knowledge of blood glucose in real-time, the participants make decisions about their meals to improve them”. So, maybe the participants could also modify their diet during the Quinoa period. How could the authors validate that the participants did not change their diet?

According to the use of nutritional pattern control where foos intake was measured using a 14-days dietary record revised corrected by a nutritionist and analyzed by the DIAL nutritional calculation program.

The authors used the FreeStyle Libre® sensor for glucose measurements. Why did they not use the indices from the sensor (time in range, GMI, CV, and others) in their statistical analysis?

Because the focus of the study was concretly the posprandial fluctuations. Despite this, it seems to us a very interesting recommendation and we will take it into account in future analyses.

Furthermore, the authors did not report MARD (Mean absolute relative difference)   during both periods of their experiments.

If your comment is according to periods V1, V2, V3: Due to the small sample size we have we used a nonparametric Wilcoxon test which uses median to compare periods rather than the mean.

On the other hand, if your comment is accordin to diets periods: We have handled glucose measurements as a function of time so we did not directly compute the change in glucose measurements, instead we used a functional linear regression model to measure the effect of diet type on the glucose functions

Overall, the study could be interesting if the authors added the above data.

Very thankful, 

All the authors.

Round 2

Reviewer 1 Report

Dear authors:

Thank you for taking into consideration my suggestions. However, I still have some questions.

  • As you state, ADA recommends two abnormal test results from the same sample or in two separate test samples to diagnose diabetes in absence of unequivocal hyperglycemia. However, diagnosis of diabetes is not only based on FPG or HbA1c. Also, you claim that “in Health Public System in Spain, HbA1c is not usually obtained from people who has not previuosly presented >126mg/dL of fasted glucose so looking at HbA1c was not an option”. But you already had an HbA1c value at V1 (Table 1, supplementary table 2), one month before starting the study intervention (This means that you performed HbA1c in all patients enrolled, so I fail to see the relevance of your previous claim). If you ALREADY had basal values of HbA1c at or above 6.5% at V1, why did you not confirm/rule out if these patients had diabetes with a second sample BEFORE V2-V3? Also, an Oral Glucose Tolerance Test  (OGTT) could have been ordered (ADA also defines as diabetes having 2-h PG >200 mg/dL (11.1 mmol/L) during OGTT), as patients with diabetes can have FPG below 126 mg/dl, and peak postprandial glucose of ≥200 mg/dl... If you did not confirm or exclude the diagnosis of diabetes in these cases, you could have enrolled patients WITH DIABETES. In short, your definition of patient with prediabetes is too lax, allowing the inclusion of patients that could very well have diabetes. This, as I have already said before, is an important flaw in your study design. You should at least exclude these patients from your study.

  • Please elaborate what kind of cardiovascular event had patient Nr. 3 as I asked previously (Stroke? AMI? PVD?). I understand that this patient had this event way before the study, but you should give more details.

Best regards

Author Response

Dear reviwer, 

First of all, we greatly appreciate the dedication and effort in reviewing our article. We proceed to reply to your letter below.

Dear authors:

Thank you for taking into consideration my suggestions. However, I still have some questions.

  • As you state, ADA recommends two abnormal test results from the same sample or in two separate test samples to diagnose diabetes in absence of unequivocal hyperglycemia. However, diagnosis of diabetes is not only based on FPG or HbA1c. Also, you claim that “in Health Public System in Spain, HbA1c is not usually obtained from people who has not previuosly presented >126mg/dL of fasted glucose so looking at HbA1c was not an option”. But you already had an HbA1c value at V1 (Table 1, supplementary table 2), one month before starting the study intervention (This means that you performed HbA1c in all patients enrolled, so I fail to see the relevance of your previous claim). If you ALREADY had basal values of HbA1c at or above 6.5% at V1, why did you not confirm/rule out if these patients had diabetes with a second sample BEFORE V2-V3? Also, an Oral Glucose Tolerance Test  (OGTT) could have been ordered (ADA also defines as diabetes having 2-h PG >200 mg/dL (11.1 mmol/L) during OGTT), as patients with diabetes can have FPG below 126 mg/dl, and peak postprandial glucose of ≥200 mg/dl... If you did not confirm or exclude the diagnosis of diabetes in these cases, you could have enrolled patients WITH DIABETES. In short, your definition of patient with prediabetes is too lax, allowing the inclusion of patients that could very well have diabetes. This, as I have already said before, is an important flaw in your study design. You should at least exclude these patients from your study.

We appreciate your response and your considerations. Our aim was not confirm type 2 diabetes diagnosis and, as we were not able to obtained HbA1c before participants were included in our study, we decided to do not repeat V1 biochemistry to confirm/refuse type 2 diabetes diagnoses. At the moment we took this decision we are not able to know if this participant presented or not diabetes in the moment of the inclusion. 

We decided to include/exclude participants according 2 of the 3 diabetes diagnosis ways and we assume that the best option would be confirm their situation with OGTT also but we were not able to do by the study characteristics and professionals researchers participating. Both the scientific committee of the hospital and the committee of experts of the Diabetes Association of Catalonia (ADC) agreed with our way of including the participants was correct.

  • Please elaborate what kind of cardiovascular event had patient Nr. 3 as I asked previously (Stroke? AMI? PVD?). I understand that this patient had this event way before the study, but you should give more details.

The participant had spontaneous atrial fibrillation that resolved on its own and did not repeat any other cardiovascular events in the years prior to the study.

Best regards

Reviewer 2 Report

The first phase of the Abstract should be revised since it is not directly supported by previous studies. Moreover, the authors did not investigate the incidence of diabetes mellitus in relation to quinoa consumption.

Similarly, the conclusions should be adapted according to the results of the present study. Speculation regarding other benefits of quinoa consumption could be included in the Discussion section.

The references for the used questionnaires for assessing physical activity and food intake should be included in the reference list.

The reference for the created quinoa products intake should be included in the reference list.

Author Response

Dear reviwer, 

First of all, we greatly appreciate the dedication and effort in reviewing our article. We proceed to reply to your letter below.

The first phase of the Abstract should be revised since it is not directly supported by previous studies. Moreover, the authors did not investigate the incidence of diabetes mellitus in relation to quinoa consumption.

Finally we have modified the section on incidence indicating the following: “Quinoa could produce a benefit on postprandial glycemia that would result in less progression to type 2 diabetes (T2D).

Similarly, the conclusions should be adapted according to the results of the present study. Speculation regarding other benefits of quinoa consumption could be included in the Discussion section.

We have modified the conclusion section indicating: “...A diet rich in quinoa reduces postprandial glycemia despite intrapersonal differences thanks to the joint action of different nutrients and the suppression of others consumed in a regular diet which supposes a brake in the progression to T2D.”

The references for the used questionnaires for assessing physical activity and food intake should be included in the reference list.

We consider it unnecessary to refer to both questions so widely used in routine clinical practice. Both are easily traceable with the indications described in Supplementary information file and in Study design section in Main document. 

The reference for the created quinoa products intake should be included in the reference list.

We have indicated: “Thus, apart from delivering quinoa, quinoa flakes and quinoa flour to the partici-pants, they were given products created with ≥70% quinoa flour and were biscuits, crackers, brioche, sponge cake, baguette bread, sliced bread and pasta (Supplementary Table 1) created and produced by Alicia-elBulli Foundation

Reviewer 3 Report

No other comments. The authors answered my comments

Author Response

Thank you.